# Revisiting XRec: How Collaborative Signals Influence LLM-Based Recommendation Explanations

**Catalin E. Brita**[*]  
*University of Amsterdam*  
*catalin.brita@student.uva.nl*

**Hieu Nguyen**[*]  
*University of Amsterdam*  
*hieu.nguyen@student.uva.nl*

**Lubov Chalakova**[*]  
*University of Amsterdam*  
*lubov.chalakova@student.uva.nl*

**Nikola Petrov**[*]  
*University of Amsterdam*  
*nikola.petrov@student.uva.nl*

**Reviewed on OpenReview:** *https://openreview.net/forum?id=cPtqOkxQqH*

## Abstract

Recommender systems help users navigate large volumes of online content by offering personalized recommendations. However, the increasing reliance on deep learning-based techniques has made these systems opaque and difficult to interpret. To address this, XRec (Ma et al., 2024) was introduced as a novel framework that integrates collaborative signals and textual descriptions of past interactions into Large Language Models (LLMs) to generate natural language explanations for recommendations. In this work, we reproduce and expand upon the findings of Ma et al. (2024). While our results validate most of the original authors' claims, we were unable to fully replicate the reported performance improvements from injecting collaborative information into every LLM attention layer, nor the claimed effects of data sparsity. Beyond replication, our contributions provide evidence that the Graph Neural Network (GNN) component does not enhance explainability. Instead, the observed performance improvement is attributed to the Collaborative Information Adapter, which can act as a form of soft prompting, efficiently encoding task-specific information. This finding aligns with prior research suggesting that lightweight adaptation mechanisms can condition frozen LLMs for specific downstream tasks. Our implementation is open-source[1].

## 1 Introduction

Recommender systems (Resnick & Varian, 1997) help users filter through vast amounts of online content by providing personalized suggestions tailored to their preferences and interests (Cheng et al., 2021; 2022). Explainable recommendation models (Zhang et al., 2020) have emerged to enhance trust, decision-making, and satisfaction by providing explanations for their suggestions (Zhang, 2019).

Recent advancements in Large Language Models (LLMs) (e.g., Touvron et al. 2023; Grattafiori et al. 2024) have revolutionized the field of explainable recommendations. Earlier work, such as PETER (Li et al., 2021), used transformers to generate explanations, while more recent works, like PEPLER (Li et al., 2023), integrated GPT-2 (Radford et al., 2019) with user-item IDs as prompts to produce contextually rich and coherent justifications. Building on these foundations, LLM2ER (Yang et al., 2024b) introduced a personalized prompt module, overcoming the limitations of relying solely on user-item IDs for explanation generation.

---

[*]Equal contribution.  
[1]https://github.com/tr2512/ReXRec

A more recent study, *XRec: Large Language Models for Explainable Recommendation* (Ma et al., 2024), introduces a novel model-agnostic approach to address the limited availability of explanation data and the poor generalization of ID-based methods in zero-shot scenarios. This framework integrates collaborative signals (patterns of user interactions, such as reviews) into LLMs by injecting them into both the input prompt and the model layers, allowing XRec to extract meaningful insights from user preferences. We chose to focus on XRec because, to our knowledge, it was one of the first methods to combine Graph Neural Networks with LLMs to generate explainable recommendations, achieving state-of-the-art performance.

This paper attempts to reproduce and extend the main findings of Ma et al. (2024) through the following:

- **Reproducing XRec results:** We use the codebase of the original paper[2] to replicate its experiments, verify the authors' claims, and analyze the required computational resources.

- **Extending the ablation study:** We investigate the role of collaborative signals by evaluating different GNN-based recommender systems that capture such signals from graph structures, including LightGCN (He et al., 2020) and NGCF (Wang et al., 2019). Additionally, we test three ablations that entirely remove graph-based collaborative information. We extend the evaluation by using Sentence Transformer Similarity (STS) (Thakur et al., 2020) and introducing LLMScore.

- **Addressing data leakage:** We identified two instances of data leakage caused by the original data generation process. The first involves using the same review to generate both the ground truth and the input to the framework, while the second is caused by the inclusion of test set information within the training data. We propose solutions to address these issues and analyze their impact.

- **Code adaptation and improvement:** We modify and improve the original codebase to support open-source LLMs for dataset generation and evaluation, facilitating future extensions and reproducibility. Additionally, we add scripts for tasks like dataset creation and data-sparsity testing.

After performing the above steps, we verified that XRec (Ma et al., 2024) outperforms the baseline models. Our findings also confirm that incorporating user and item profile information improves performance and that both the profile and the injection component can contribute to improved personalization. However, we encountered challenges in reproducing the claims that injecting collaborative information across all LLM attention layers improves performance and that XRec is more effective under increased data sparsity.

Furthermore, our in-depth analysis of the impact of collaborative information reveals that the performance improvement in XRec (Ma et al., 2024) can be attributed to the Collaborative Information Adapter. We hypothesize this occurs because (1) collaborative signals may be redundant when unique identification features are available, even if randomly assigned, since the adapter can learn these signals as effectively as GNN-based models and (2) the adapter functions as a form of soft prompting, efficiently encoding task-specific information (Lester et al., 2021; Li et al., 2021).

While reproducing the datasets proposed by Ma et al. (2024), we identified two instances of data leakage, which we subsequently addressed in our attempt to reproduce the datasets. Despite the presence of these data leakage cases, our observations indicate that they did not significantly affect the overall claims or results.

## 2 Scope of reproducibility

Ma et al. (2024) introduced XRec, a post-hoc technique that improves recommender system explainability by providing user-level personalization. Ma et al. (2024) make five key claims, which we investigate:

- **Claim 1: XRec improves over baselines in explainability and stability.** The model improves explanations in recommender systems by integrating collaborative information and textual descriptions into LLMs, outperforming state-of-the-art baselines in both explainability and stability.

---

[2]https://github.com/HKUDS/XRec

- **Claim 2: Collaborative information injection improves explainability and stability.** XRec achieves higher accuracy by injecting collaborative information across all attention layers of the LLM, rather than simply passing it in the input prompt.

- **Claim 3: User and item profiles improve explainability and stability.** Adding user and item profiles to the prompt improves explainability and stability by providing contextual information.

- **Claim 4: XRec has superior performance with increasing sparsity.** The model demonstrates robust performance across varying sparsity levels, showing improved results as user frequency decreases. Moreover, in zero-shot scenarios, its performance is comparable to other sparsity levels.

- **Claim 5: The model generates personalized explanations.** XRec generates customized explanations for each distinct user-item interaction, offering users meaningful and unique insights.

Moreover, we extend the work by answering the following research questions:

- **RQ 1:** How do Graph Neural Network (GNN)-based collaborative signals and the Collaborative Information Adapter influence the explainability and stability of XRec?

- **RQ 2:** What are the effects of dataset reproduction in XRec on explainability and stability, especially when addressing sources of data leakage?

## 3 Methodology

### 3.1 Model description

XRec (Ma et al., 2024) is a framework that uses a Large Language Model (LLM) to generate natural language explanations of why a recommender system suggested an item to a user. XRec (Ma et al., 2024), as visualized in Figure 1, is a framework that uses an LLM to generate natural-language explanations for recommendations. It consists of four main components: (i) a Graph Neural Network (GNN)-based recommender system that learns user-item interaction patterns, (ii) an adapter that transforms these learned embeddings into the LLM's input space, (iii) textual profiles of users and items constructed via summarization, and (iv) an attention injection mechanism that reinforces collaborative signals during explanation generation.

**GNN-based recommender system** The GNN-based recommender system models user-item interactions as a graph, with users and items as nodes and interactions as edges. GNNs embed users and items into fixed-size vectors using a message-passing mechanism (Gilmer et al., 2017), which aggregates information from neighboring nodes. XRec employs LightGCN (He et al., 2020), a model with a simple message-passing mechanism which lacks linear transformations and nonlinear activations during aggregation. This is shown by Eq. 1, where $\mathbf{e}_x^{(l)}$ is the embedding of node $x$ at layer $l$, and $\mathcal{N}_x$ is the set of neighbors of node $x$.

$$\mathbf{e}_u^{(l+1)} = \sum_{i \in \mathcal{N}_u} \frac{1}{\sqrt{|\mathcal{N}_u|}\sqrt{|\mathcal{N}_i|}} \mathbf{e}_i^{(l)} \tag{1}$$

To investigate how the GNN affects XRec's performance, we experiment by replacing LightGCN with NGCF (Wang et al., 2019). NGCF has a more complex message-passing mechanism that uses nonlinear activations and learnable weight matrices $(\mathbf{W}_1, \mathbf{W}_2)$, as shown in Eq. 2.

$$\mathbf{e}_u^{(l+1)} = \sigma \left( \mathbf{W}_1 \mathbf{e}_u^{(l)} + \sum_{i \in \mathcal{N}_u} \frac{1}{\sqrt{|\mathcal{N}_u|\,|\mathcal{N}_i|}} \left( \mathbf{W}_1 \mathbf{e}_i^{(l)} + \mathbf{W}_2 \left( \mathbf{e}_i^{(l)} \odot \mathbf{e}_u^{(l)} \right) \right) \right) \tag{2}$$

Both GNNs are trained with the Bayesian Personalized Ranking loss (Rendle et al., 2009) (Eq. 3), which favors higher scores for observed interactions. $\ell_2$ regularization is also applied. Here, $L$ denotes the loss

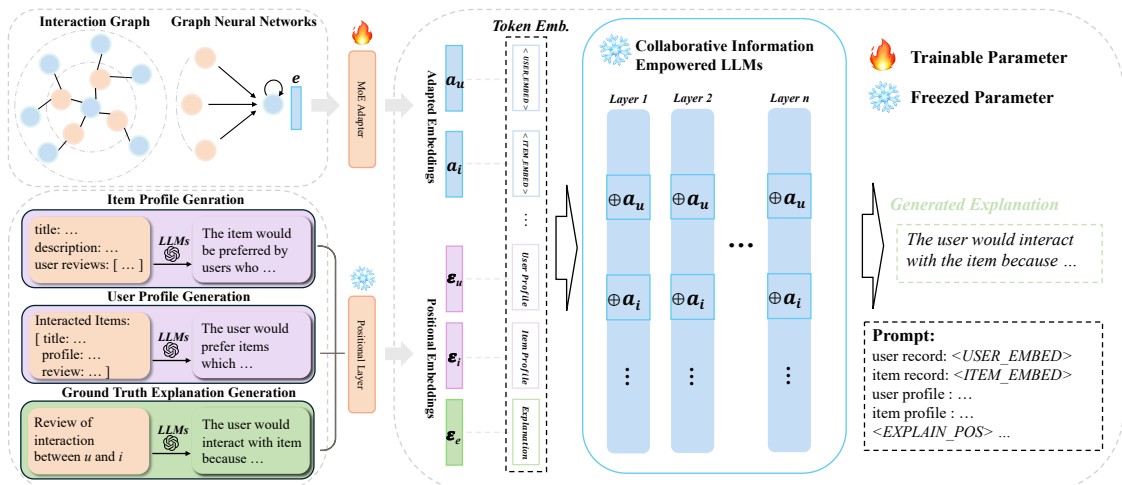

Figure 1: XRec framework consisting of (i) **GNN-based recommender system** (top left): embeds user-item interactions. (ii) **Collaborative Information Adapter** (MoE Adapter): transforms user and item embeddings to LLM input space. (iii) **User and Item Text-Based Profiles** (bottom left): text profile of users and items (iv) **Attention Injection Mechanism**: injects collaborative information into each LLM attention layer. Embeddings and profiles are combined into a prompt that is processed by the LLM, where collaborative signals are used throughout to generate the final explanation. Reprinted from Ma et al. (2024).

function, M is the number of users, $\hat{y}_{ui}$ refers to the predicted rating of user $u$ for item $i$, $\sigma$ represents the sigmoid function, $\mathbf{E}^{(0)}$ denotes the embedding matrix, and $\lambda$ is the $\ell_2$ regularization parameter.

$$L = -\sum_{u=1}^{M} \sum_{i \in \mathcal{N}_u} \sum_{j \notin \mathcal{N}_u} \ln \sigma(\hat{y}_{ui} - \hat{y}_{uj}) + \lambda \|\mathbf{E}^{(0)}\|^2 \tag{3}$$

**Collaborative Information Adapter**  To effectively incorporate the collaborative information learned by the recommender system into the LLM, XRec uses an adapter which transforms the user and item embeddings into the input space of the LLM. This adapter uses a Mixture-of-Experts (MoE) architecture which effectively captures diverse aspects of the collaborative information.

**User and Item Text-Based Profiles**  In addition to the adapted GNN embeddings, XRec takes as input textual profiles of users and items. Item profiles are created by prompting an LLM to summarize each item's description, while user profiles are generated by summarizing descriptions of items the user has interacted with. XRec's input is constructed as shown in Figure 1, where <USER_EMBED> and <ITEM_EMBED> are the embeddings generated by the GNN.

**Attention Injection Mechanism**  A potential issue with prompt construction is that collaborative information, which is crucial for capturing user-item relationships, is limited to a single token. To ensure the collaborative signal is not lost in a long prompt, it is injected into the key, query, and value of all self-attention layers. This is shown in Eq. 4, where $\mathbf{W}$ is the projection matrix, $\mathbf{a}_i$ represents the adapted user or item embedding, and $f_{\{q,k,v\}}^{\text{original}}$ and $f_{\{q,k,v\}}^{\text{new}}$ denote the query, key, and value before and after injection, respectively.

$$f_{\{q,k,v\}}^{\text{new}}(\mathbf{x}_i) = f_{\{q,k,v\}}^{\text{original}}(\mathbf{x}_i) + \mathbf{W}_{\{q,k,v\}} \cdot \mathbf{a}_i \tag{4}$$

## 3.2  Evaluation metrics

We evaluate the model using LLM-based metrics, as they better reflect human judgment and capture context-aware semantic similarity, rather than relying solely on token-level analysis. First, to ensure a fair comparison

with the baseline model performance reported by Ma et al. (2024), we use **GPTScore** (Wang et al., 2023). GPTScore uses an LLM as a judge by prompting GPT-3.5 Turbo to assess the semantic similarity between ground-truth explanations and generated explanations. Wang et al. (2023) showed that GPTScore has a high correlation with human judgment and shows a deep semantic understanding, making it suitable for assessing the alignment between generated explanations and groundtruth explanations. However, GPTScore's use of a closed-source model limits accessibility and reproducibility and introduces model-specific bias.

To address these limitations, we propose **LLMScore**, which uses multiple open-source LLMs as a jury (Verga et al., 2024), each outputting an alignment score with the same prompt used by GPTScore. LLMScore averages these scores to mitigate model-specific biases and improve accessibility. Scores are obtained over five runs to ensure stability. The models used are *LLaMA 3.1 8B Instruct*, *LLaMA 3.2 3B Instruct* (Grattafiori et al., 2024), *Gemma 2 9B-IT* (Rivière et al., 2024), and *Qwen 2.5 7B-Instruct* (Yang et al., 2024a). The prompt used for scoring is provided in Appendix B.

In addition to LLMScore, we use several other metrics to capture explanation quality. **BERTScore** (Zhang et al., 2019) computes cosine similarity over BERT (Kenton & Toutanova, 2019) embeddings for token-level comparison of synonyms and paraphrases. Ma et al. (2024) reports Precision, Recall, and F1, but we use only F1 since it combines the other two. Precision and Recall details are reported in Appendix E.2. Complementing this, **BARTScore** (Yuan et al., 2021) measures the likelihood of regenerating ground-truth text using pre-trained BART (Lewis, 2019), making it effective for assessing fluency and coherence. To further align with human judgment, we include **BLEURT** (Sellam et al., 2020), a fine-tuned BERT model trained on human-annotated text pairs, which enhances sensitivity to subtle differences in meaning.

Beyond semantic quality, we assess personalization by measuring the diversity of generated explanations. We use **USR (Unique Sentence Ratio)** (Li et al., 2020) - the ratio of unique to total explanations, providing insight into the variability of generated content. To build on this, we use **STS (Sentence Transformer Similarity)** (Thakur et al., 2020), which uses SBERT (Reimers, 2019) embeddings and cosine similarity to evaluate the semantic similarity of generated explanations.

Finally, to compare the performance of LightGCN (He et al., 2020) and NGCF (Wang et al., 2019), we employ four metrics: **Precision@20** measures the proportion of relevant items among the top 20 recommendations, indicating accuracy; **Recall@20** measures the proportion of relevant items retrieved, indicating coverage; **NDCG@20 (Normalized Discounted Cumulative Gain)** measures ranking quality by weighing relevance and position, prioritizing relevant items appearing earlier and **MRR@20 (Mean Reciprocal Rank)** focuses on the rank of the first relevant item, rewarding higher-ranked relevant items.

## 4 Results

### 4.1 Datasets

**Dataset processing**  To build XRec, Ma et al. (2024) use three datasets, each containing user reviews for various items: **Amazon Review Data**[3] (Ni et al., 2019), **Google Local Data**[4] (Li et al., 2022; Yan et al., 2023), and **Yelp Open Dataset**[5]. These were processed into what we refer to as **[Og]Amazon**, **[Og]Google**, and **[Og]Yelp** datasets. As we were unable to find the exact procedure used to create the datasets, we independently reproduce them and refer to them as **[Re]Amazon**, **[Re]Google**, and **[Re]Yelp**. Detailed processing steps for these datasets, along with information about data splitting, can be found in Appendix C. Table 1 presents a statistical comparison between the original and reproduced datasets.

**Train-test overlap**  XRec is trained in two phases: (1) training the recommender system using the full interaction dataset, and (2) training the Collaborative Information Adapter using the explanation dataset (a subset of the interaction dataset). We observed overlap between the train, validation, and test splits of the original explanation and interaction datasets proposed in Ma et al. (2024), with overlap percentages provided in Appendix C.3. This issue is addressed in the reproduced datasets.

---

[3] https://cseweb.ucsd.edu/~jmcauley/datasets/amazon_v2/
[4] https://jiachengli1995.github.io/google/index.html
[5] https://business.yelp.com/data/resources/open-dataset/

Table 1: Dataset statistics from Ma et al. (2024) ([Og]) and our reproduced version ([Re]).

| Dataset | [Og]Amazon | [Re]Amazon | [Og]Google | [Re]Google | [Og]Yelp | [Re]Yelp |
|---|---|---|---|---|---|---|
| #Users | 15,349 | 15,069 | 22,582 | 19,503 | 15,942 | 15,962 |
| #Items | 15,247 | 15,028 | 16,557 | 18,998 | 14,085 | 14,085 |
| #Interactions | 360,839 | 350,644 | 411,840 | 400,038 | 393,680 | 393,680 |

**Ground-truth and profile generation**  For the original datasets, ground truth explanations are generated based on user reviews in order to make user preferences explicit. For item profiles, the Google and Amazon datasets use item names and descriptions, while the Yelp dataset also includes all user reviews of that item. User profiles are generated based on information from each item a user has interacted with, including the item's name, its associated profile, and the review the user left for that item. To improve dataset reproducibility, we use LLaMA 3.1 8B Instruct (Grattafiori et al., 2024) for generation, replacing the closed-source GPT model used in Ma et al. (2024).

**Input contamination**  The user generation process introduces dataset leakage across all original datasets, while the Yelp item profile generation process introduces additional leakage specific to Yelp. This occurs because user reviews are used to generate both the ground-truth explanations and the input (profiles). This raises two issues: (1) It creates an overlap between the input and ground truth, potentially causing the model to rely on this overlap instead of learning meaningful patterns from user-item interactions, which limits generalization during testing as this shortcut is unavailable (2) It compromises the evaluation process by leaking ground-truth test information into the test input profile. To address this in our reproduced datasets, we exclude all review information from input prompts used to generate profiles.

## 4.2  Training Configurations

The training process consists of two phases: training the GNN model and training the Collaborative Information Adapter module. The training process followed the hyperparameter configurations specified in Ma et al. (2024). An early stopping criterion is applied to the GNN-based models based on the Recall@20 metric. The LLM model used is LLaMA 2 7B (Touvron et al., 2023), which remains frozen throughout the training process. Appendix A shows the exact hyperparameter values for each phase.

Training and inference are conducted using a single NVIDIA A100 40GB GPU, while evaluation is performed on two NVIDIA T4 16GB GPUs. Training takes approximately 15 hours for the Amazon and Google datasets and 12 hours for the Yelp dataset, for both the original and reproduced versions. Inference requires about 4 hours per dataset, and evaluation takes around 1.5 hours per dataset.

## 4.3  Results reproducing original paper

This section details our reproduction of the findings of Ma et al. (2024), where we retrain the XRec model and its ablations. Our goal is to validate key claims regarding the model's explainability and stability, ensuring consistency with the original results.

**Claim 1:** *XRec improves over baselines in explainability and stability.* We reproduce the results of Ma et al. (2024) by retraining (1) XRec on the original datasets, and (2) XRec (w/o Profile), where user and item profiles were excluded from the input prompts to ensure a fair comparison with the baselines that did not have access to profile information. Table 2 presents the original XRec (Ma et al., 2024) results alongside the scores from our reproduced models. Our reproduction shows an improvement in GPTScore and a decrease in $GPT_{std}$ across all datasets. While we cannot determine the cause of this change, because the metric relies on a closed-source model[6], we observe that the performance trends remain consistent with the original results.

For BERT-based metrics, BARTScore, and BLEURT, the results remain consistent with the original trends, with minor negligible variations. These metrics confirm that XRec maintains high explainability and stability,

---

[6]This prevents us from verifying whether the same model version and internal system prompt were used in both cases.

Table 2: Comparison of baseline models and our reproduced results. Values for all baselines and XRec (denoted [Orig]XRec) are from (Ma et al., 2024), while our reproduced XRec results appear in *italics*. BERT$^{F1}$ aggregates Precision and Recall, which are reported in Appendix E.2. **Bold** indicates the highest score; underlined indicates the second highest.

| Metrics | Explainability ↑ | | | | | Stability ↓ | | | |
|---|---|---|---|---|---|---|---|---|---|
| | GPTScore | BERT$^{F_1}$ | BARTScore | BLEURT | USR | GPT$_{std}$ | BERT$^{F_1}_{std}$ | BART$_{std}$ | BLEURT$_{std}$ |
| **[Og]Amazon** | | | | | | | | | |
| Att2Seq$^\dagger$ | 76.08 | 0.3687 | -3.9440 | -0.3302 | 0.7757 | 12.56 | 0.1275 | 0.5080 | 0.299 |
| NRT$^\dagger$ | 75.63 | 0.3443 | -3.9806 | -0.4073 | 0.5413 | 12.82 | 0.1321 | 0.5101 | 0.3104 |
| PETER$^\dagger$ | 77.65 | 0.4043 | -3.8968 | -0.2937 | 0.8480 | 11.21 | 0.1098 | 0.5144 | 0.2667 |
| PETER+$^\dagger$ | 76.07 | 0.3876 | -3.9647 | -0.3293 | 0.4493 | 11.99 | 0.1245 | 0.5131 | 0.2805 |
| PEPLER$^\dagger$ | 78.77 | 0.3543 | -3.9142 | -0.2950 | 0.9563 | 11.38 | 0.0893 | 0.5064 | 0.2195 |
| [Orig]XRec$^\dagger$ | 82.57 | **0.4122** | -3.8035 | **-0.1061** | 1.0000 | 9.60 | 0.0800 | 0.4832 | **0.1780** |
| [Orig]XRec (w/o profile)$^\dagger$ | 81.77 | 0.4106 | -3.8218 | -0.1294 | 1.0000 | 9.60 | 0.0786 | **0.4799** | 0.1803 |
| *XRec* | ***83.75*** | *0.4095* | ***-3.7773*** | *-0.1793* | *1.0000* | *9.40* | *0.0817* | *0.4985* | *0.2256* |
| *XRec (w/o profile)* | *83.28* | *0.4094* | *-3.8328* | *-0.1839* | *1.0000* | *9.16* | *0.0772* | *0.4801* | *0.2178* |
| **[Og]Yelp** | | | | | | | | | |
| Att2Seq$^\dagger$ | 63.91 | 0.2379 | -4.5316 | -0.6707 | 0.7583 | 15.62 | 0.1147 | 0.5616 | 0.247 |
| NRT$^\dagger$ | 61.94 | 0.1495 | -4.6142 | -0.7913 | 0.2677 | 16.81 | 0.1581 | **0.5612** | 0.2728 |
| PETER$^\dagger$ | 67.00 | 0.2513 | -4.4100 | -0.5816 | 0.8750 | 15.57 | 0.2230 | 0.5800 | 0.3555 |
| PETER+$^\dagger$ | 67.98 | 0.2833 | -4.3973 | -0.5355 | 0.8637 | 13.80 | 0.1701 | 0.5665 | 0.3421 |
| PEPLER$^\dagger$ | 67.54 | 0.3052 | -4.4563 | -0.3354 | 0.9143 | 14.18 | 0.1050 | 0.5777 | 0.2524 |
| [Orig]XRec$^\dagger$ | 74.53 | **0.3730** | -4.3911 | **-0.2287** | 1.0000 | 11.45 | **0.0852** | 0.5770 | 0.2322 |
| [Orig]XRec (w/o profile)$^\dagger$ | 71.81 | 0.3657 | -4.4035 | -0.2486 | 1.0000 | 12.71 | 0.0919 | 0.5717 | **0.2272** |
| *XRec* | ***79.72*** | *0.3527* | ***-4.2969*** | *-0.3089* | *1.0000* | ***10.65*** | *0.0967* | *0.5758* | *0.2334* |
| *XRec (w/o profile)* | *75.68* | *0.3636* | *-4.3717* | *-0.3303* | *1.0000* | *11.13* | *0.1014* | *0.5676* | *0.2514* |
| **[Og]Google** | | | | | | | | | |
| Att2Seq$^\dagger$ | 61.31 | 0.3636 | -4.2627 | -0.4671 | 0.5070 | 17.47 | 0.1403 | 0.6663 | 0.3198 |
| NRT$^\dagger$ | 58.27 | 0.3496 | -4.2915 | -0.4838 | 0.2533 | 19.16 | 0.1571 | 0.6620 | 0.3118 |
| PETER$^\dagger$ | 65.16 | 0.3881 | -4.1527 | -0.3375 | 0.4757 | 17.00 | 0.2005 | 0.6701 | 0.3272 |
| PETER+$^\dagger$ | 66.74 | 0.4047 | -4.1273 | -0.3467 | 0.4887 | 15.23 | 0.1411 | 0.6515 | 0.3095 |
| PEPLER$^\dagger$ | 69.71 | 0.3987 | -4.1542 | -0.2047 | 0.8660 | 17.11 | 0.1353 | 0.6800 | 0.3114 |
| [Orig]XRec$^\dagger$ | 69.12 | 0.4311 | -4.1647 | -0.2437 | 0.9993 | 14.24 | 0.0937 | **0.5700** | **0.2114** |
| [Orig]XRec (w/o profile)$^\dagger$ | 69.71 | 0.4310 | **-4.1142** | -0.2026 | 0.997 | 14.09 | 0.1034 | 0.6465 | 0.2439 |
| *XRec* | *71.13* | *0.4331* | *-4.1686* | *-0.2366* | *0.9993* | *13.22* | *0.0935* | *0.6560* | *0.2509* |
| *XRec (w/o profile)* | ***71.66*** | ***0.4395*** | *-4.1237* | ***-0.1829*** | *1.000* | *13.05* | *0.0931* | *0.6529* | *0.2392* |

† This score was taken from Ma et al. (2024).

as the best and second-best scores are consistently achieved by our reproduced models. Additionally, while XRec (w/o Profile) performs slightly worse than XRec, it still consistently outperforms the baselines. This suggests that even when restricted to the same level of information as the baselines, XRec has superior performance. Overall, these results support **Claim 1**.

Table 3: Comparison of GPTScore and LLMScore for XRec and XRec (w/o profile) on the original datasets, with LLMScore averaged over five runs. Given their high correlation, only LLMScore is used in later analyses.

| Dataset | Model | GPTScore | LLMScore |
|---|---|---|---|
| [Og]Amazon | XRec | 83.75 | $67.53 \pm 11.65$ |
| | XRec (w/o profile) | 83.28 | $66.95 \pm 11.65$ |
| [Og]Yelp | XRec | 79.72 | $61.67 \pm 12.61$ |
| | XRec (w/o profile) | 75.68 | $59.82 \pm 12.11$ |
| [Og]Google | XRec | 71.13 | $55.81 \pm 13.04$ |
| | XRec (w/o profile) | 71.66 | $56.49 \pm 12.98$ |

To verify **Claim 1**, we use GPTScore for a fair comparison with the baselines from Ma et al. (2024). However, reproducing results is challenging since GPTScore is closed-source. Instead, we propose LLMScore, which employs multiple open-source language models for scoring. Table 3 shows that LLMScore mirrors GPTScore's

patterns: best performance on Amazon, worst on Google, and reduced performance without profiles except on Google. Therefore, LLMScore is the primary evaluation metric for the remaining experiments.

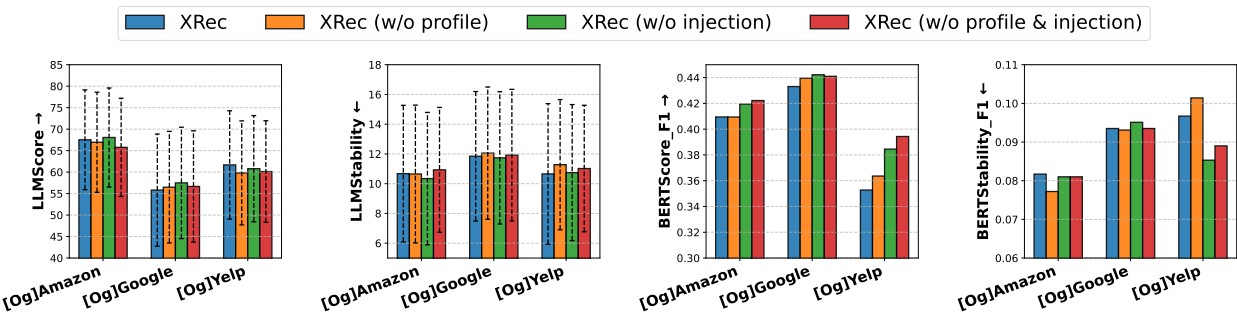

Figure 2: Ablation study results of the original datasets on the full model and three variants: without profile, without injection and without both. The vertical bars represent the standard deviation obtained over 5 runs. $BERT^{F_1}$ is deterministic and is shown without error bars.

**Claim 2:** *Collaborative information injection improves explainability and stability.* To verify this claim, we perform ablation studies using: (1) XRec, (2) XRec (w/o profile), where user and item profiles are not given as input, (3) XRec (w/o injection), where embeddings are not injected into the attention layers, and (4) XRec (w/o profile & injection). Following Ma et al. (2024), we evaluate them using LLMScore and $BERT^{F_1}$.

Figure 2 shows that on LLMScore, XRec follows a similar trend as reported by Ma et al. (2024), outperforming the other configurations on Yelp and ranking second on Amazon. However, on the Google dataset, XRec performs the worst. Although there is high variance in LLMScore, we include an explanation in Appendix E.1 and show that all scoring LLMs exhibit the same trend, confirming the conclusiveness of these results.

Moreover, in terms of $BERT^{F_1}$, XRec consistently ranks last or ties for the worst performance across all three datasets. On the other hand, for LLMScore and LLMStability, XRec (w/o injection) outperforms all other ablations on two datasets (Amazon and Google), and ranks second on the other dataset (Yelp). These results contradict **Claim 2** and suggest that, on average, injection reduces performance.

One possible explanation for the lack of performance improvement from injecting collaborative information lies in the characteristics of the underlying graph neural network (GNN). According to Qin et al. (2024), injection is most effective when the model can disentangle the various latent factors that shape user preferences and item attributes. However, LightGCN (He et al., 2020) lacks this property, which may explain its negative impact on performance.

**Claim 3:** *User and item profiles improve explainability and stability.* We evaluate this claim with the ablation study in Figure 2. XRec (w/o profile) performs worse than XRec in terms of LLMScore and LLMStability on Amazon and Yelp. Similarly, XRec (w/o injection & profile) underperforms across all three datasets compared to XRec (w/o injection). However, models without a profile exhibit a slight yet negligible improvement in $BERT^{F_1}$ compared to those that incorporate profiles. These findings support **Claim 3**.

**Claim 4:** *XRec has superior performance with increasing sparsity.* We investigate the model's performance under varying levels of data sparsity. Specifically, as done in Ma et al. (2024), we divided the test data into six groups: one for zero-shot users (users absent from the training data) and five quantiles representing increasing ranges of user interactions. Appendix C.4 provides interaction frequency ranges and statistics for each group. As shown in Figure 3, this grouping reveals that while BERTScore shows a slight improvement for users with sparser interactions, the LLM-based metrics behave differently: explainability decreases and stability increases with growing sparsity. These conflicting trends suggest that **Claim 4** is not reproducible and needs further investigation, though performance generally improves with higher interaction frequency.

**Claim 5:** *The model generates personalized explanations.* Ma et al. (2024) measure personalization using USR (Li et al., 2020), which captures lexical uniqueness but overlooks semantic equivalence. For instance, "I am going to the store" and "I am heading to the store" both score 1.0, despite being semantically identical.

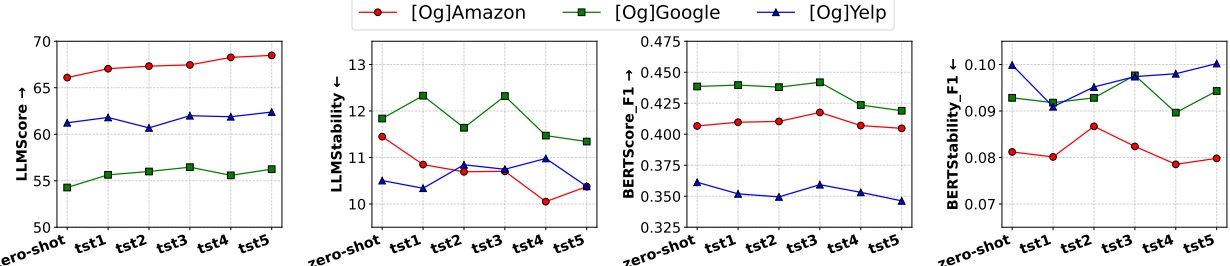

Figure 3: XRec's performance on six data splits with increasing sparsity level on the original datasets.

To test this, we sampled user-item interactions from each dataset and generated 100 explanations per sample. In all cases, USR was 1.0 even though the explanations conveyed the same meaning. This suggests that USR is not an appropriate measure of personalization. To address this, we compute an aggregate STS score by averaging the pairwise STS values (Thakur et al., 2020), providing an estimate of the overall personalization of generated explanations. On the same explanations, STS yields an average score of 0.78, indicating that the explanations are highly similar in meaning despite minor lexical differences. However, STS tends to assign higher similarity scores to sentences with similar structures, which is often the case with our generated explanations. Despite this limitation, STS provides a more meaningful measure of similarity than USR.

Table 4: Evaluation results of different XRec variations across datasets on the STS metric.

| Dataset | XRec | XRec (w/o profile) | XRec (w/o injection) | XRec (w/o profile & injection) | XRec (w/o GNN) |
|---|---|---|---|---|---|
| [Og]Amazon | 0.5667 | 0.6563 | 0.6614 | 0.6936 | 0.4139 |
| [Og]Yelp | 0.3802 | 0.3601 | 0.3671 | 0.4149 | 0.2759 |
| [Og]Google | 0.5891 | 0.5910 | 0.6006 | 0.5863 | 0.2807 |

Table 4 presents the average pairwise STS scores for each dataset. The [Og]Amazon and [Og]Google explanations exhibit overall high STS, suggesting a lower degree of personalization in comparison to the explanations for [Og]Yelp. This difference can be attributed to the higher level of detail in Yelp reviews compared to the other datasets. This shows that data quality affects the personalization capabilities of XRec. Additionally, we observe that for [Og]Amazon, incorporating user-profiles and injection improves personalization. However, for [Og]Google, these factors do not significantly impact the STS score. Finally, we notice that removing the GNN makes generated explanations more diverse. We hypothesize that this is due to the fact that when XRec is trained with the GNN component, the MoE adapter learns to prompt the model in such a way that the explanations follow a similar structure. As a result, removing this component allows the model to produce more diverse explanations, but at the cost of reduced explainability scores as shown in Table 2.

### 4.4 Results beyond original paper

This section explores additional aspects of XRec's performance. We analyze the impact of GNN collaborative signals and the MoE adapter, and assess the effect of data leakage mitigation. These extensions provide deeper insights into the model's capabilities and potential limitations.

#### 4.4.1 Impact of GNN and Collaborative Information Adapter

**RQ 1:** *How do Graph Neural Network (GNN)-based collaborative signals and the Collaborative Information Adapter influence the explainability and stability of XRec?* To assess the impact of GNN-based collaborative signals on XRec, we replace its originally used LightGCN (He et al., 2020) with NGCF (Wang et al., 2019), referring to this variant as XRec w/ NGCF. For clarity, we refer to XRec with its original GNN architecture as XRec w/ LightGCN in this section. NGCF (Wang et al., 2019) is selected due to its similar architecture to LightGCN (He et al., 2020), with only additional linear transformation and non-linear activation. This allows us to investigate the relationship between GNN recommendation performance and XRec explainability

while minimizing confounding factors from architectures. Additionally, we test three collaborative-signal ablations: XRec (w/o GNN, fixed), where GNN outputs are replaced with fixed random embeddings, XRec (w/o GNN, random), where these are replaced with variable random embeddings, and XRec (w/o GNN & MoE), removing both the GNN and the Collaborative Information Adapter (MoE).

Table 5: Performance comparison of LightGCN and NGCF models. The best values are in **bold**.

| Dataset | Model | Recall | NDCG | Precision | MRR |
|---|---|---|---|---|---|
| [Og]Amazon | LightGCN | **0.0719** | **0.0426** | **0.0175** | **0.0724** |
| | NGCF | 0.0514 | 0.0319 | 0.0131 | 0.0587 |
| [Og]Yelp | LightGCN | **0.0614** | **0.0382** | **0.0160** | **0.0677** |
| | NGCF | 0.0571 | 0.0365 | 0.0156 | 0.0668 |
| [Og]Google | LightGCN | **0.0791** | **0.0433** | **0.0129** | **0.0575** |
| | NGCF | 0.0571 | 0.0327 | 0.0094 | 0.0455 |

We compare the performance of the two recommender systems on the item recommendation task. Table 5 shows that LightGCN outperforms NGCF across all evaluation metrics and datasets. We use the embeddings from these two models as input to the MoE Adapter. Table 6 shows that XRec (w/ NGCF) outperforms XRec (w/ LightGCN) on average, achieving higher BERT-based scores on two datasets and a higher LLMScore on one. This suggests that better item recommendations do not guarantee improved XRec explainability.

Table 6: Performance comparison of XRec with LightGCN and NGCF as collaborative information algorithms, as well as three variants where explicit collaborative information is removed. $\text{BERT}^{F1}$ aggregates Precision and Recall, which are reported in Appendix E.2. Values in **bold** indicate the highest score, while underlined values denote the second-best. LLMScore and LLMStability are averaged over five runs.

| Metrics | Explainability ↑ | | | | | Stability ↓ | | | |
|---|---|---|---|---|---|---|---|---|---|
| | LLMScore | $\text{BERT}^{F1}$ | BARTScore | BLEURT | STS | LLMStability | $\text{BERT}^{F1}_{std}$ | $\text{BART}_{std}$ | $\text{BLEURT}_{std}$ |
| **[Og]Amazon** | | | | | | | | | |
| XRec (w/ LightGCN) | $67.53 \pm 11.61$ | 0.4095 | **-3.7773** | -0.1783 | 0.57 | $10.67 \pm 4.59$ | 0.0817 | 0.4985 | 0.2256 |
| XRec (w/ NGCF) | $67.97 \pm 11.66$ | **0.4164** | -3.7789 | -0.1445 | 0.62 | $10.58 \pm 4.57$ | 0.0806 | 0.4913 | 0.2118 |
| XRec (w/o GNN, fixed) | **$68.05 \pm 11.69$** | 0.4089 | -3.8171 | -0.1461 | 0.67 | **$10.19 \pm 4.65$** | **0.0764** | 0.4766 | 0.2077 |
| XRec (w/o GNN, random) | $65.33 \pm 11.81$ | 0.4179 | -3.8575 | **-0.1264** | 0.69 | $11.31 \pm 4.38$ | 0.0816 | 0.4861 | **0.1990** |
| XRec (w/o GNN & MoE) | $62.66 \pm 12.14$ | 0.3013 | -4.0502 | -0.4729 | **0.42** | $11.05 \pm 4.29$ | 0.0804 | **0.4336** | 0.2439 |
| **[Og]Yelp** | | | | | | | | | |
| XRec (w/ LightGCN) | **$61.67 \pm 12.61$** | 0.3527 | -4.2969 | -0.3089 | 0.38 | $10.65 \pm 4.73$ | 0.0967 | 0.5758 | 0.2334 |
| XRec (w/ NGCF) | $60.95 \pm 12.49$ | 0.3736 | **-4.2719** | -0.3053 | 0.33 | $10.67 \pm 4.76$ | **0.0841** | 0.5657 | 0.2128 |
| XRec (w/o GNN, fixed) | $61.20 \pm 12.49$ | 0.3658 | -4.3070 | -0.2971 | 0.36 | **$10.61 \pm 4.78$** | 0.0897 | 0.5694 | **0.2104** |
| XRec (w/o GNN, random) | $59.34 \pm 12.01$ | **0.3926** | -4.4425 | **-0.2608** | 0.45 | $11.00 \pm 4.46$ | 0.0877 | 0.5790 | 0.2112 |
| XRec (w/o GNN & MoE) | $58.68 \pm 13.56$ | 0.2710 | -4.6124 | -0.3615 | **0.27** | $11.51 \pm 4.80$ | 0.0953 | **0.5392** | 0.2174 |
| **[Og]Google** | | | | | | | | | |
| XRec (w/ LightGCN) | $55.81 \pm 13.05$ | **0.4331** | -4.1686 | -0.2366 | 0.59 | **$11.84 \pm 4.35$** | 0.0935 | 0.6560 | 0.2509 |
| XRec (w/ NGCF) | $55.04 \pm 13.29$ | 0.4233 | -4.2445 | -0.2652 | 0.54 | $12.12 \pm 4.30$ | 0.0946 | 0.6558 | 0.2635 |
| XRec (w/o GNN, fixed) | **$57.25 \pm 12.87$** | 0.4306 | **-4.1158** | **-0.1776** | 0.53 | $12.04 \pm 4.44$ | 0.0959 | 0.6488 | 0.2398 |
| XRec (w/o GNN, random) | $56.71 \pm 12.89$ | 0.4310 | -4.1595 | -0.1780 | 0.63 | $11.96 \pm 4.45$ | 0.0937 | 0.6542 | **0.2390** |
| XRec (w/o GNN & MoE) | $50.40 \pm 14.29$ | 0.2456 | -4.5504 | -0.5188 | **0.28** | $12.17 \pm 4.58$ | **0.0918** | **0.5725** | 0.2451 |

We observe that removing both the collaborative information and the adapter in XRec (w/o GNN & MoE) results in the lowest performance, highlighting the importance of these components. However, this boosts personalization as measured by STS.

When comparing with XRec (w/o GNN, fixed), which uses a random embedding vector instead of embeddings generated by GNNs, we do not observe any performance drop. Instead, we find that it achieves the best performance on two datasets and ranks second-best on the other. Meanwhile, XRec (w/o GNN, random) performs worse than models with the MoE adapter but better than those without it. This suggests that removing collaborative signals from the GNN has minimal impact on explainability and that the MoE effectively learns meaningful information, though it struggles when the input is random rather than fixed.

We hypothesize that this effect arises from two possible causes, though further research is needed to confirm the exact mechanism. First, the MoE adapter may interpret the embedding vector as a set of unique

identifiers, enabling it to implicitly learn collaborative signals even without GNN-based embeddings. The poorer performance of XRec (w/o GNN, random) supports this hypothesis, as the randomized inputs disrupt the model's ability to retain meaningful identification information. Second, both STS scores and manual inspection revealed structural differences in sentence composition between models with and without the MoE adapter. This suggests that incorporating the lightweight adapter, positioned almost as a prefix to the prompt, may unintentionally act as a form of soft prompting. This effect aligns with prior work in prompt tuning Lester et al. (2021); Li et al. (2021), which demonstrated that small, lightweight adapters can condition LLMs for downstream tasks without requiring full fine-tuning.

### 4.4.2 Effect of Data Leakage Mitigation

**RQ 2:** *What are the effects of dataset reproduction in XRec on explainability and stability, especially when addressing train-test overlap and the use of shared reviews for both profile generation and ground truth creation?* We identified two cases of data leakage in the original datasets (Ma et al., 2024): **Input Contamination**, where the same reviews are used to generate both ground-truth explanations and input profiles; and **Train-Test Leakage**, where training, validation, and test sets overlap. To assess the impact of the leakage, we train XRec on the original datasets and our reproduced versions, where the issues are mitigated.

Table 7: XRec performance on Original ([Og]) datasets with data leakage and Reproduced ([Re]) datasets with mitigated leakage. $BERT^{F1}$ aggregates BERTScore Precision and Recall, which are reported in Appendix E.2. The best values are in **bold**. LLMScore and LLMStability are averaged over five runs.

| Metrics | Explainability ↑ | | | | | Stability ↓ | | | |
|---|---|---|---|---|---|---|---|---|---|
| | LLMScore | $BERT^{F_1}$ | BARTScore | BLEURT | STS | LLMStability | $BERT^{F_1}_{std}$ | $BART_{std}$ | $BLEURT_{std}$ |
| [Og]Amazon | **67.53 ± 11.61** | 0.4095 | **-3.7773** | -0.1793 | **0.5667** | **10.67 ± 4.59** | **0.0817** | **0.4985** | **0.2256** |
| [Re]Amazon | 64.80 ± 11.25 | **0.4396** | -3.8135 | **-0.1474** | 0.7489 | 11.84 ± 3.99 | 0.0943 | 0.5583 | 0.2599 |
| [Og]Yelp | 61.67 ± 12.61 | 0.3527 | -4.2969 | -0.3089 | **0.3802** | **10.65 ± 4.73** | 0.0967 | **0.5758** | 0.2334 |
| [Re]Yelp | **62.75 ± 11.89** | **0.4300** | **-4.0250** | **-0.1765** | 0.5186 | 10.78 ± 4.52 | **0.0937** | 0.5813 | **0.2176** |
| [Og]Google | 55.81 ± 13.05 | 0.4331 | -4.1686 | -0.2366 | **0.5891** | **11.84 ± 4.35** | **0.0935** | 0.6560 | 0.2509 |
| [Re]Google | **59.72 ± 12.45** | **0.4349** | **-3.9298** | **-0.1283** | 0.6074 | 12.57 ± 4.43 | 0.0960 | **0.6089** | **0.2451** |

Table 7 demonstrates that mitigating data leakage improves the explainability of XRec, as reflected in higher BERTScore and BLEURT values. This result is counterintuitive since input contamination typically inflates performance by providing the model with direct access to ground-truth information. However, we cannot definitely attribute these improvements solely on data leakage mitigation, as other factors—such as variations in selected reviews, or differences in the models used to generate the ground truth and explanations—may have steered the results in this direction. Additionally, Figure 4 shows a similar data sparsity trend as Figure 3, indicating that data leakage does not impact the model's behaviour across sparsity levels. The consistency between these results indicates that mitigating leakage does not compromise the validity of earlier findings.

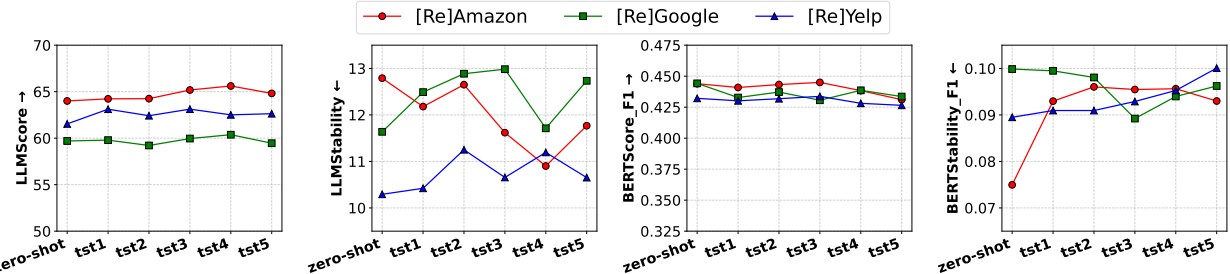

Figure 4: XRec's performance on six data splits with increasing sparsity level on the reproduced datasets.

# 5   Conclusion

Our reproduction study and analysis of XRec (Ma et al., 2024) confirm that the model improves over baselines in both explainability and stability. Although our results verify the impact of user and item profiles, they do not support the claim that collaborative information injection through the layers of the LLM improves explainability and stability, as XRec without injection performs better on two out of three datasets. Additionally, we find that explainability and stability improve with more information about the specific user, contradicting the claim that XRec performs better with increasing data sparsity.

Additional experiments show that removing the GNN-based recommendation system has a minimal impact on model explainability, as performance remains stable or slightly improves without it. We hypothesize that this occurs because (1) when distinct identification features of users and items are present, even if randomly assigned, the MoE can effectively capture collaborative patterns without relying on a GNN-based model, making those signals potentially redundant, and (2) the MoE adapter acts similarly to soft prompting, embedding task-specific knowledge in a way that improves explainability (Lester et al., 2021; Li et al., 2021).

**Limitations and Future Work**   Our experiments show performance gains after removing data leakage, but these cannot be conclusively attributed to leakage alone, as factors such as review selection and model differences can also play a role. While the results suggest data leakage may affect performance, its exact impact remains unclear. Future work should explore this through additional experiments.

Another limitation is in the personalization metrics. USR focuses on uniqueness, which may miss personalization quality, while STS can favor similar structures over truly personalized meaning. These issues limit conclusions about personalization effectiveness. Future work should develop metrics that better capture meaningful personalization beyond surface-level similarity.

Finally, while we hypothesize that the MoE acts as soft prompting, this may be amplified by metrics favoring structural similarity over meaning. Future work should examine the MoE's impact on explanation structure and meaning, and develop more robust evaluation metrics.

We believe future work on explainable recommendations can build on our study by improving the evaluation metrics. We recommend adopting LLMScore as the primary evaluation metric to mitigate language models' inherent bias toward favoring their own generated text or models consistent with them (Panickssery et al., 2024). To reduce variation in scores, evaluations should use a limited rating scale (e.g., 0-5) and include in-context examples based on human-annotated judgments (Stureborg et al., 2024). Alternatively, an LLM could act as a discriminator in pairwise comparisons (Wu et al., 2024), choosing the better of two generated explanations against a ground-truth reference. Lastly, we encourage future work to perform data generation and evaluation using distinct models due to the same self-consistency issue (Panickssery et al., 2024).

**What was easy:**   Ma et al. (2024) provided most of the model implementation with documentation on running the training process. The model architecture description was detailed and easy to follow. Additionally, the authors provided evidence linked to specific claims, simplifying the reproduction process.

**What was difficult:**   Some parts of this study were more challenging than expected. The dataset preprocessing code was incomplete, making it difficult to rebuild the full pipeline. Ablation and evaluation scripts were also not provided, and documentation lacked details on metrics and dataset-specific settings.

**Communication with original authors:**   They promptly responded to our questions and clarified details about the data generation process and the hyperparameters we could alter to replicate their datasets.

### Acknowledgments

We thank Emmanouil Georgios Lionis for their feedback and inspiration, and Clara Rus for their insightful feedback on this manuscript, which significantly contributed to the success of this project.

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

## A  Hyperparameters

The models were trained and evaluated using the parameters listed in Table 8:

Table 8: Hyperparameter configurations for the GNN-based recommender system and the Information Collaboration Adapter module (MoE), both optimized using Adam (Kingma & Ba, 2015).

| Hyperparameter | GNN | MoE |
|---|---|---|
| Batch size | 1024 | 1 |
| Number of epochs | 300 | 1 |
| Optimizer | Adam | Adam |
| Learning rate | 0.001 | 0.001 |
| Number of layers | 4 | - |
| Embedding size | 64 | - |
| Early Stopping Patience | 10 epochs | - |
| Number of experts | - | 8 |
| Dropout rate | - | 0.2 |
| Gating Router Noise Factor | - | 0.01 |

## B  GPTScore Prompt

To acquire a GPTScore and LLMScore for each XRec explanation we use the following prompt proposed by Ma et al. (2024):

**System Prompt:**

```
Score the given explanation against the ground truth on a scale from 0 to 100,
focusing on the alignment of meanings rather than the formatting.
Provide your score as a number and do not provide any other text.
```

**Prompt:**

```
{
    "prediction" : <XREC explanation>,
    "reference"  : <ground truth explanation>
}
```

# C  Datasets

## C.1  Dataset Processing

To evaluate XRec, Ma et al. (2024) use three datasets from distinct domains, which they subsequently process. The first dataset, **Amazon Review Data**[7], proposed by Ni et al. (2019), includes reviews and product metadata, where only the book subset is used for XRec. The second dataset, **Google Local Data**[8], introduced by Li et al. (2022); Yan et al. (2023), contains Google Maps reviews and business metadata, where only the California subset is used. Finally, the **Yelp Open Dataset**[9], includes reviews and business metadata, and the entire dataset serves as input.

Ma et al. (2024) provide example processing code for the **Yelp Open Dataset**. Their approach involves:

1. Filtering out items without a name, description, or categories;

2. Filtering out all reviews with a rating of 3 or less;

3. Randomly subsampling the remaining users;

4. Retaining only interactions within the 8-core interaction graph[10].

We follow these steps to reproduce the datasets, which we refer to as [Re]Amazon, [Re]Google, and [Re]Yelp to distinguish them from the original Amazon, Google, and Yelp datasets in XRec's repository. However, whereas Ma et al. (2024) use the full Google dataset and work with a manually extracted 8-core subgraph, we use the smaller 10-core version to reduce computational overhead. For consistency, we process the Amazon and Yelp datasets into 10-core versions.

The ground-truth explanations are generated from a subset of interactions, taken from the pre-processed dataset, following the original procedure in Ma et al. (2024). For the Amazon and Google datasets, we include reviews with more than 10 words in order to reduce noise.

## C.2  Data Splitting

We provide the Train, Validation, and Test splits for the reproduced interaction datasets—[Re]Amazon, [Re]Google, and [Re]Yelp—in Table 9. These interaction datasets are used to train the Recommender System. From each interaction dataset, we extract a subset containing interactions with ground truth explanations, which we refer to as the explanation dataset. This explanation dataset is used to train the XRec framework. To ensure consistency, the splits for the explanation dataset were designed to match the numbers of the corresponding original datasets (Ma et al., 2024), and the same split ratios were applied to the interaction datasets.

Table 9: Train, Validation, and Test splits for [Re]Amazon, [Re]Google, and [Re]Yelp datasets.

| Split | [Re]Amazon | | [Re]Google | | [Re]Yelp | |
|---|---|---|---|---|---|---|
| | Explanations | Interactions | Explanations | Interactions | Explanations | Interactions |
| Train | 95,841 | 303,246 | 94,663 | 345,846 | 74,212 | 337,797 |
| Validation | 11,980 | 37,905 | 11,833 | 43,231 | 9,277 | 42,227 |
| Test | 3,000 | 9,493 | 3,000 | 10,961 | 3,000 | 13,656 |
| Sum | 110,821 | 350,644 | 109,496 | 400,038 | 86,489 | 393,680 |

---

[7]https://cseweb.ucsd.edu/~jmcauley/datasets/amazon_v2/

[8]https://jiachengli1995.github.io/google/index.html

[9]https://business.yelp.com/data/resources/open-dataset/

[10]A "k-core" of a graph is a subgraph in which every node has a degree of at least k, meaning each node is connected to at least k other nodes. For example, in an 8-core graph, every user or item must have at least 8 interactions to be included.

### C.3 Overlap Between Original Dataset Splits

Ma et al. (2024) train their system using two datasets: (1) the full interaction dataset (for training the recommendation system) and (2) the explanation dataset (a subset of the interaction dataset, used to train the Collaborative Information Adapter). The explanation dataset includes textual explanations of reviews corresponding to specific interactions. However, we observed that during the splitting of these datasets, overlaps occasionally occur, leading to instances where data from one dataset is exposed to the other. In this section, we present the percentage of overlap between the train, validation, and test splits of the explanation and interaction datasets. Table 10 summarize these overlaps for the [Og]Amazon, [Og]Google, and [Og]Yelp datasets introduced by Ma et al. (2024).

Table 10: Percentage of overlap between the splits of Explanations and Interactions for [Og]Amazon, [Og]Google and [Og]Yelp datasets.

| Dataset | Expl. split | Interactions (%) | | |
|---|---|---|---|---|
| | | **Train** | **Validation** | **Test** |
| Amazon | Train | 79 | 10 | 10 |
| | Validation | 79 | 10 | 9 |
| | Test | 79 | 9 | 10 |
| Google | Train | 80 | 9 | 10 |
| | Validation | 80 | 9 | 9 |
| | Test | 78 | 10 | 10 |
| Yelp | Train | 79 | 10 | 10 |
| | Validation | 80 | 10 | 9 |
| | Test | 81 | 9 | 9 |

### C.4 Data sparsity statistics

Figure 5 provides the statistics of all sparsity sets used to reproduce **Claim 4**.

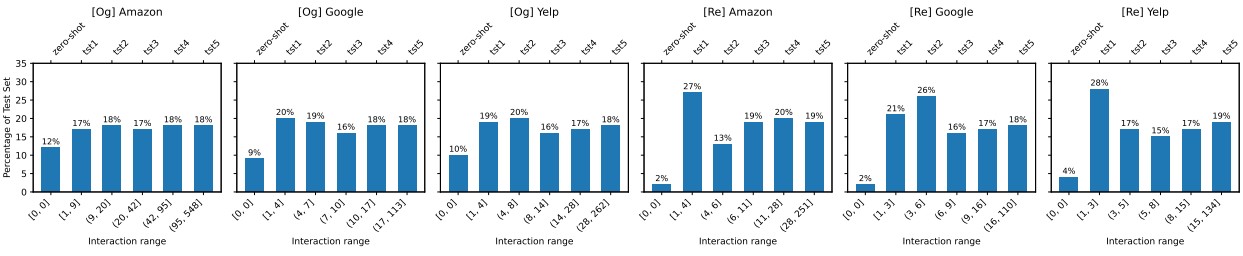

Figure 5: Data sparsity set statistics including user interaction frequency intervals and the percentage of data allocated to each sparsity set from the test set consisting of 3000 interactions. Each panel corresponds to one dataset and displays the proportion of interactions in each sparsity set (zero-shot, tst1–tst5) for both original ([Og]) and reproduced ([Re]) datasets. The bottom x-axis shows the exact frequency intervals, while the top x-axis indicates sparsity-set names.

## D Carbon Emissions

We estimate the carbon emissions from the experiments using the $CO_2e = CI \times PUE \times P \times t$ formula, where the carbon intensity (CI) in the Netherlands is 0.370 $CO_2e$ emissions per kWh. The power usage effectiveness (PUE) values for the different setups are 1.2 and 1.1. The power consumption is 0.25 kW for A100 GPUs and 0.07 kW for T4 GPUs, with runtime durations of 280 hours and 160 hours, respectively. Based on these calculations, the estimated carbon emissions for the experiments amount to approximately 31 kg $CO_2e$ and 4.5 kg $CO_2e$. In total, the experiments resulted in approximately 35.5 kg $CO_2e$ emissions.

# E    Extended Results

## E.1    Ablation Results per LLM Judge

The error bars are computed by first obtaining five stochastic predictions from the LLM for each individual test instance, calculating a within-instance standard deviation ($\sigma_i$), and subsequently averaging these $\sigma_i$ across the dataset. While informative for capturing prediction-level noise, this procedure inadvertently inflated uncertainty estimates at the dataset level. The inflation occurs for two primary reasons: (1) each $\sigma_i$, computed from only five samples, inherently possesses significant sampling error; and (2) averaging $\sigma_i$ across all instances preserves prediction-level variability without benefiting from the typical $\sqrt{n}$ reduction that occurs when averaging metrics at the corpus level.

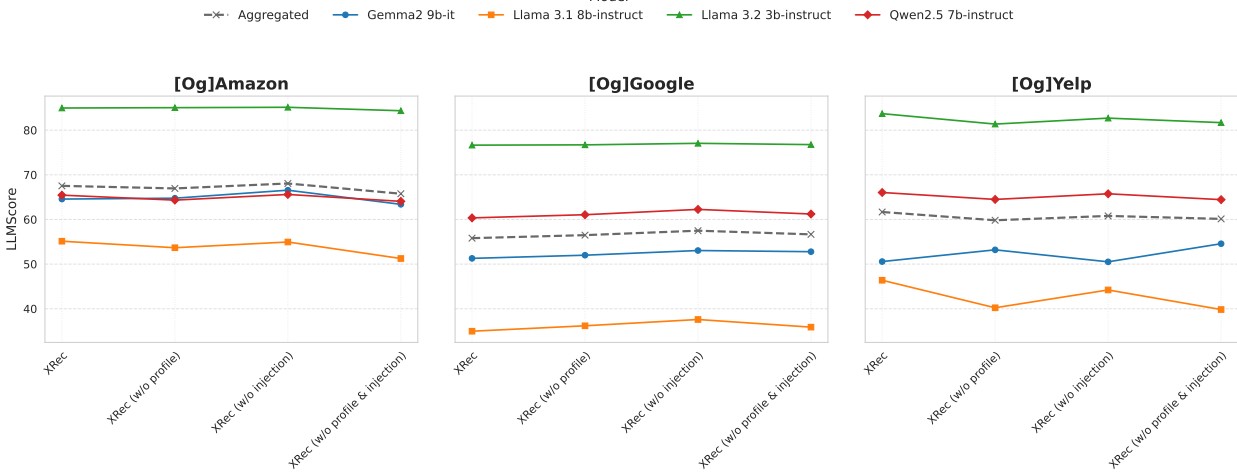

Figure 6: Scores from individual LLM judges (colored lines) evaluating XRec explanations across ablations on the [Og]Amazon, [Og]Google, and [Og]Yelp datasets. The average score across these judges ("Aggregated" dashed line) represents the final LLMScore reported in Figure 2. All LLMs show agreement on the [Og]Amazon and [Og]Google datasets, while for the [Og]Yelp dataset, all LLMs agreed except for the Gemma2 model.

## E.2    BERTScore Precision and Recall

In this section, we present the extended results with BERT Precision (Zhang et al., 2019)($BERT^p$), BERT Recall (Zhang et al., 2019) ($BERT^R$) metric. Given the predicted sentence BERT representation is $\hat{x} = [\hat{x}_1, \hat{x}_2, ...\hat{x}_n]$ and the reference sentence representation is $x = [x_1, x_2, ...x_m]$. BERT Precision, BERT Recall and BERT F1 is defined as in Equations 5, 6 and 7. The detailed BERT Precision and Recall are presented in Tables 11, 12, and 13.

$$BERT^R = \frac{1}{n} \sum_{x_i \in x} \max_{\hat{x}_j \in \hat{x}} \mathbf{x}_i^\top \hat{\mathbf{x}}_j \tag{5}$$

$$BERT^P = \frac{1}{m} \sum_{\hat{x}_j \in \hat{x}} \max_{x_i \in x} \mathbf{x}_i^\top \hat{\mathbf{x}}_j \tag{6}$$

$$BERT^{F1} = 2\frac{BERT^P \cdot BERT^R}{BERT^P + BERT^R} \tag{7}$$

Table 11: Comparison of baseline models and our reproduced results for BERTScore Precision ($\text{BERT}^P$) and Recall ($\text{BERT}^R$). Baseline and original XRec results (denoted as [Orig]XRec) are taken from (Ma et al., 2024), while our reproduced results (XRec) are shown in *italics*. **Bold** indicates the highest score, and underlined indicates the second highest. Table 2 aggregates these results under $\text{BERT}^{F1}$ (F1 Score).

| Metrics | Explainability ↑ | | Stability ↓ | |
|---|---|---|---|---|
| | $\text{BERT}^P$ | $\text{BERT}^R$ | $\text{BERT}^P_{std}$ | $\text{BERT}^R_{std}$ |
| **[Og]Amazon** | | | | |
| Att2Seq[†] | 0.3746 | 0.3624 | 0.1691 | 0.1051 |
| NRT[†] | 0.3444 | 0.3440 | 0.1804 | 0.1035 |
| PETER[†] | **0.4279** | 0.3799 | 0.1334 | 0.1035 |
| PETER+[†] | 0.4119 | 0.3626 | 0.1576 | 0.1077 |
| PEPLER[†] | 0.3506 | 0.3569 | 0.1105 | 0.0935 |
| [Orig]XRec[†] | 0.4193 | **0.4038** | 0.0836 | 0.0920 |
| [Orig]XRec (w/o profile)[†] | 0.4194 | 0.4004 | 0.0819 | 0.0955 |
| *XRec* | *0.4139* | *0.4037* | *0.0863* | *0.0933* |
| *XRec (w/o profile)* | *0.4207* | *0.3967* | ***0.0802*** | ***0.0888*** |
| **[Og]Yelp** | | | | |
| Att2Seq[†] | 0.2099 | 0.2658 | 0.1583 | 0.1074 |
| NRT[†] | 0.0795 | 0.2225 | 0.2293 | 0.1134 |
| PETER[†] | 0.2102 | 0.2983 | 0.3315 | 0.1298 |
| PETER+[†] | 0.2594 | 0.3097 | 0.2522 | 0.1174 |
| PEPLER[†] | 0.2920 | 0.3183 | 0.1476 | **0.1044** |
| [Orig]XRec[†] | **0.3946** | **0.3506** | **0.0969** | 0.1048 |
| [Orig]XRec (w/o profile)[†] | 0.3879 | 0.3427 | 0.1087 | 0.1072 |
| *XRec* | *0.3541* | *0.3504* | *0.1126* | *0.1108* |
| *XRec (w/o profile)* | *0.3814* | *0.3451* | *0.1218* | *0.1110* |
| **[Og]Google** | | | | |
| Att2Seq[†] | 0.3619 | 0.3653 | 0.1855 | 0.1247 |
| NRT[†] | 0.3509 | 0.3495 | 0.2176 | 0.1267 |
| PETER[†] | 0.3892 | 0.3905 | 0.2819 | 0.1356 |
| PETER+[†] | 0.4125 | 0.3975 | 0.1893 | 0.1244 |
| PEPLER[†] | 0.3806 | 0.4093 | 0.1602 | 0.1154 |
| [Orig]XRec[†] | 0.4546 | 0.4069 | **0.0972** | **0.1031** |
| [Orig]XRec (w/o profile)[†] | 0.4427 | 0.4187 | 0.1180 | 0.1171 |
| *XRec* | ***0.4632*** | *0.4025* | *0.0986* | *0.1163* |
| *XRec (w/o profile)* | *0.4594* | ***0.4191*** | *0.1001* | *0.1150* |

† This score was taken from Ma et al. (2024).

Table 12: BERTScore Precision (BERT$^P$) and Recall (BERT$^R$) comparison of XRec with LightGCN and NGCF as collaborative information algorithms, as well as three variants where explicit collaborative information is removed. Values in **bold** indicate the highest score, while underlined values denote the second-best. Table 6 aggregates these results under BERT$^{F1}$ (F1 Score).

| Metrics | Explainability ↑ | | Stability ↓ | |
|---|---|---|---|---|
| | BERT$^P$ | BERT$^R$ | BERT$^P_{std}$ | BERT$^R_{std}$ |
| **[Og]Amazon** | | | | |
| XRec (w/ LightGCN) | 0.4139 | 0.4037 | 0.0863 | 0.0933 |
| XRec (w/ NGCF) | 0.4277 | **0.4088** | 0.0825 | 0.0934 |
| XRec (w/o GNN, fixed) | 0.4200 | 0.3963 | **0.0077** | 0.0899 |
| XRec (w/o GNN, random) | **0.4361** | 0.3984 | 0.0842 | 0.0940 |
| XRec (w/o GNN & MoE) | 0.2909 | 0.3102 | 0.1007 | **0.0796** |
| **[Og]Yelp** | | | | |
| XRec (w/ LightGCN) | 0.3541 | 0.3504 | 0.1126 | 0.1108 |
| XRec (w/ NGCF) | 0.3828 | **0.3635** | **0.0953** | **0.1047** |
| XRec (w/o GNN, fixed) | 0.3737 | 0.3572 | 0.1103 | 0.1039 |
| XRec (w/o GNN, random) | **0.4376** | 0.3476 | 0.0967 | 0.1098 |
| XRec (w/o GNN & MoE) | 0.2796 | 0.2609 | 0.1045 | 0.1088 |
| **[Og]Google** | | | | |
| XRec (w/ LightGCN) | **0.4632** | 0.4025 | 0.0986 | 0.1163 |
| XRec (w/ NGCF) | 0.4517 | 0.3943 | 0.1027 | 0.1136 |
| XRec (w/o GNN, fixed) | 0.4414 | **0.4192** | 0.1081 | 0.1141 |
| XRec (w/o GNN, random) | 0.4520 | 0.4094 | **0.0958** | 0.1187 |
| XRec (w/o GNN & MoE) | 0.2094 | 0.2806 | 0.1049 | **0.1019** |

Table 13: BERTScore Precision (BERT$^P$) and Recall (BERT$^R$) comparison of XRec on (1) Original ([Og]) datasets with data leakage and (2) Reproduced ([Re]) datasets with mitigated leakage. The best values are in **bold**. Table 7 aggregates these results under BERT$^{F1}$ (F1 Score).

| Metrics | Explainability ↑ | | Stability ↓ | |
|---|---|---|---|---|
| | BERT$^P$ | BERT$^R$ | BERT$^P_{std}$ | BERT$^R_{std}$ |
| [Og]Amazon | 0.4139 | 0.4037 | **0.0863** | **0.0933** |
| [Re]Amazon | **0.4563** | **0.4218** | 0.0986 | 0.1049 |
| [Og]Yelp | 0.3541 | 0.3504 | 0.1126 | 0.1108 |
| [Re]Yelp | **0.4382** | **0.4207** | **0.1002** | **0.1076** |
| [Og]Google | **0.4632** | 0.4025 | **0.0986** | 0.1163 |
| [Re]Google | 0.4450 | **0.4238** | 0.1004 | **0.1107** |

