# OpenReview forum: "Revisiting XRec: How Collaborative Signals Influence LLM-Based Recommendation Explanations"
_TMLR — Accepted by TMLR_

### Review · Reviewer_PzX6 · 2025-03-19

**Summary Of Contributions:**

Explainable recommendation systems are widely studied today as essential technologies. The method at the core of this paper, XRec, uses a large-language model  (LLM)-based approach proposed by Ma et al. in 2024. The original XRec generates collaborative signals using user interactions such as reviews and injects these into the language model to effectively model user preferences for recommendations.

Beyond the above original work, the authors revisited XRec, reproducing its implementation and experiments to compare performance. The replication study added ablation experiments not conducted in the original XRec paper and considered data leakage issues that emerged during reproduction. The results confirmed the value of LLM-based explainable recommendation systems and provide deeper insights into the effectiveness of the GNN component previously assumed in existing research.

**Audience:**

Yes

**Broader Impact Concerns:**

Not applicable to this paper in my opinion.

**Claims And Evidence:**

Yes

**Requested Changes:**

**[Critical to securing my recommendation for acceptance]**

While I respect the authors' contribution to reproducibility, the paper contains numerous underexplained points, likely due to page limitations. In its current state, I cannot confidently recommend acceptance.

Below are my concerns regarding explanations, experimental result reliability, and presentation:
- The paper cites Figure 1 from XRec, but inadequately explains its individual components. This creates a gap in demonstrating the approach's value and undermines trust in the work. The selection of XRec appears arbitrary, the GPTScore explanation is insufficient (leaving its measurement purpose unclear), and the choice of NGCF lacks motivation—all of which weaken the overall contribution. Most critically, the unexplained selection of NGCF and the minimal explanation of GPTScore present significant concerns.
- As highlighted in [W2], the tables' current size and density impair readability. This makes it impossible to directly verify the consistency of results claimed in Claim 1, diminishing the paper's persuasiveness.
- The paper needs to explain the vertical bars in Claim 2, as Fig. 3's current description lacks conviction. For example, while it discusses the inverse relationship between increasing LLM Score and decreasing BERT Score F1, insufficient explanations of Fig. 2 and GPTScore make it unclear whether this represents a consistent pattern or mere coincidence. The observed stability issues suggest the latter. Furthermore, the paper should justify its transition to using only F1—if Recall/Precision are not necessary, simplifying this throughout could strengthen the paper's arguments.

**[Comments to strengthen the work in my view]**
- As noted in [W5], I am unsure whether discussing CO2e is necessary, especially at the cost of omitting important technical explanations.
- For Table 13, visualizations (e.g., using histograms) could be helpful to show the division is valid and reliable.

**Strengths And Weaknesses:**

Strengths:

- [S1] The paper is generally well-explained, clearly stating its claims (Claim1-Claim5) and research questions (RQ1-RQ2).
- [S2] The study significantly focuses on reproducibility, which is crucial in recommendation system research (e.g., the well-known RecSys’19 paper), and provides ample appendices and information.
- [S3] The paper addresses the often overlooked aspect of data processing reproducibility, identifying and resolving issues.

Weaknesses (See also Requested Changes)

- [W1] There are several underexplained points and concepts, possibly due to trying to fit the content within 12 pages.
- [W2] The readability of tables and figures has decreased, likely due to space constraints within 12 pages.
- [W3] Explanations of terms like GPTScore and LLMScore are insufficient for researchers unfamiliar with TMLR or RS backgrounds.
- [W4] The reasoning behind choosing NGCF for the ablation study is unclear.
- [W5] It is uncertain whether the discussion on CO2e is necessary, especially if it leads to dropping other explanations.
- [W6] Related to [W2], there are difficulties in quickly understanding the consistency of scores and the assertions of claims from the tables and figures; improving the explanation and presentation would be beneficial.

---

> ### Author Response · Authors · 2025-04-30
> **Review response**
>
> We thank the reviewer for the detailed reviews. We have addressed your concerns and requested changes and updated the manuscript accordingly. The changes in the manuscript are highlighted with teal-colored text.
>
> - The paper cites Figure 1 from XRec, but inadequately explains its individual components.
>
> We have added a more detailed explanation of XRec’s overall architecture and the roles of its individual components, as depicted in Figure 1. The changes are located at the start of Section 3.
>
> - The selection of XRec appears arbitrary
>
> We thank the reviewer for the comment. At the time this project began, XRec was the state-of-the-art (SOTA) in explainable recommendations. Furthermore, it was one of the few works that integrated both traditional Graph Neural Networks (GNNs) and Large Language Models (LLMs) to generate explanations for recommendations. We believe that reproducing XRec provides the research community with valuable insights into the combined effects of traditional GNN methodologies with modern LLMs for the specific task of explainable recommendation. We have added this justification to the Introduction section.
>
> - The GPTScore explanation is insufficient (leaving its measurement purpose unclear)
>
> We thank the reviewer for pointing this out. We have updated the manuscript with a more detailed explanation of GPTScore in Section 3.2: GPTScore is a metric that uses Large Language Models to evaluate natural language generation by prompting them to rate text quality against groundtruth texts. The score has been shown to correlate highly with human judgements, making it a suitable choice for explainable recommendations where the generated explanations need to align with human intent and understanding.
>
> - the choice of NGCF lacks motivation
>
> We chose NGCF for the ablation study because it shares a very similar architecture with LightGCN (the base GNN used in XRec). LightGCN can be understood as a simplified version of NGCF, specifically lacking the feature transformation matrices and non-linear activations present in NGCF's propagation layers. To investigate the relationship between GNN performance on the recommendation task and XRec performance on explainability, we wanted to select a comparison GNN with minimal architectural changes compared to LightGCN. This approach helps to avoid confounding factors that an entirely different GNN architecture might introduce. Therefore, NGCF appeared to be a suitable choice. We have updated Section 4.4.1 of the manuscript with this reasoning.
>
> - The paper needs to explain the vertical bars in Claim 2, as Fig. 3's current description lacks conviction. For example, while it discusses the inverse relationship between increasing LLM Score and decreasing BERT Score F1, insufficient explanations of Fig. 2 and GPTScore make it unclear whether this represents a consistent pattern or mere coincidence. The observed stability issues suggest the latter.
>
> We have added a description to the caption of Figure 2 explaining that the vertical bars represent the score variance (e.g., standard deviation) across 5 runs. To demonstrate that the score pattern is consistent rather than coincidental, we have included Figure 6 in Appendix E.1, which shows the scores of individual LLMs used within XRec. The figure reveals that across the different ablation settings, four LLMs exhibit the same trend on [Og]Amazon and [Og]Google, and three out of four have the same trend on [Og]Yelp, confirming that this is a consistent pattern despite high variance.
>
> - Furthermore, the paper should justify its transition to using only F1—if Recall/Precision are not necessary, simplifying this throughout could strengthen the paper's arguments.
>
> We thank the reviewer for this insightful comment. We agree that for the main presentation of results, the F1 score serves as an effective summary metric, combining both Precision and Recall. Therefore, we have revised the main text to use the F1 for reproducibility. Comprehensive Precision and Recall results are now detailed in Appendix E.2 for completeness.
>
> - the tables' current size and density impair readability
>
>  As mentioned above, we have removed Precision and Recall scores from the main results tables in the body of the paper and moved them to the appendix. This simplification significantly improves the readability of the tables.
>
> - As noted in [W5], I am unsure whether discussing CO2e is necessary, especially at the cost of omitting important technical explanations.
>
> We moved the discussion about CO2 emissions to the appendix.
>
> - For Table 13, visualizations (e.g., using histograms) could be helpful to show the division is valid and reliable.
>
> We updated the divisions with histograms in Figure 5 to show that the division across different levels of sparsity is valid.

---

### Review · Reviewer_776v · 2025-03-29

**Summary Of Contributions:**

Recommender systems, especially those using deep learning, have become complex and hard to interpret. XRec (Ma et al., 2024) was introduced to address this by integrating collaborative signals and textual descriptions into Large Language Models (LLMs) for generating natural language explanations for recommendations. This paper aims to reproduce and expand on the findings of Ma et al. (2024). The authors replicated XRec experiments, extended the ablation study, addressed data leakage issues in the original data generation, and adapted the codebase. Their results validated that XRec outperforms baseline models and that incorporating user and item profile information improves performance. However, they could not fully replicate claims about injecting collaborative information into every LLM attention layer, improving performance and the effects of data sparsity. Their analysis showed that the Collaborative Information Adapter, rather than the Graph Neural Network (GNN) component, was responsible for the performance improvement, acting as a form of soft prompting. The paper concludes by highlighting areas for future research, such as developing better evaluation methods for personalization and examining the influence of the adapter on explanation structure and meaning.

**Audience:**

Yes

**Broader Impact Concerns:**

No potential negative societal impacts are explicitly discussed, which is suggested to be included.

**Claims And Evidence:**

Yes

**Requested Changes:**

Please kindly refer to the Weaknesses for potential requested changes.

**Strengths And Weaknesses:**

Strengths:
* The paper conducts a comprehensive reproduction of the XRec study. By replicating experiments, extending ablation studies, and addressing data leakage, it provides a detailed and systematic examination of the original work.
* The discovery that the Collaborative Information Adapter is the main driver of performance improvement, rather than the GNN component as might have been initially assumed, is a significant contribution. This finding not only clarifies the inner workings of XRec but also aligns with prior research on lightweight adaptation mechanisms for LLMs.
* Making their implementation open-source enables other researchers to build on this work, further validate the findings, and explore new directions in explainable recommendation systems.

Weaknesses:
* While the findings regarding reproduction are interesting and the conclusion deviates from the original paper, the paper focuses solely on validating and supplementing XRec without proposing new methods, which could make the paper less significant.
* While the authors challenge the claims that injecting collaborative information across all LLM
attention layers improves performance, and XRec is more effective under increased data sparsity, could the authors also provide some insights on how to improve XRec to improve the performance on sparse data and mitigate the complexity of injecting collaborative information in all attention layers?
* For the new observations such as the extending of the ablation study, could they also be able to generalize to other similar models or different datasets outside of those used in the original study.

---

> ### Author Response · Authors · 2025-04-30
> **Review response**
>
> We thank the reviewer for the positive words. Our answers to the three mentioned weaknesses are as follows:
>
> ## [W1] While the findings regarding reproduction are interesting and the conclusion deviates from the original paper, the paper focuses solely on validating and supplementing XRec without proposing new methods, which could make the paper less significant.
>
> We understand the concern that only validating an existing method does not seem novel, but we believe that our reproduction and analysis offer significant contributions:
>
> - **Identifying the performance drivers**: Our experiments show that, contrary to the original claims, the Collaborative Information Adapter, rather than the GNN component, is the main contributor to XRec's performance gains. Moreover, we notice that injecting collaborative signals into every LLM attention layer can slightly reduce performance. These findings indicate that the pipeline proposed by Ma et. al. (2024) can be simplified, reducing computational overhead without compromising effectiveness. By highlighting which components are essential and which are not, our work can help future research focus on more important components.
>
> - **Clarifying sparsity limitations**: Our study also shows that XRec does not perform better under increased data sparsity, contrary to the original claim. This finding aligns with the intuition that having more data about a user generally improves recommendation quality.
>
> - **Identified and mitigated data leakage**: While reproducing the work, we found and addressed two instances of data leakage. We believe that this is valuable to other researchers using the same training and evaluation pipeline, as it helps ensure fair comparisons and more reliable conclusions in follow-up studies.
>
> We hope that this deeper understanding of XRec, along with the insights uncovered in our reproduction study, will help guide the development of more efficient and robust explainable recommendation systems.
>
> ## [W2] While the authors challenge the claims that injecting collaborative information across all LLM attention layers improves performance, and XRec is more effective under increased data sparsity, could the authors also provide some insights on how to improve XRec to improve the performance on sparse data and mitigate the complexity of injecting collaborative information in all attention layers?
>
> We believe the claim that XRec performs better under increased sparsity is counterintuitive, as having more user data generally improves performance, a trend our results support. We leave improving performance under sparsity to future work, though we expect the same relative trend to hold, with lower performance at higher sparsity levels. Regarding collaborative information injection, we suggest that the limited expressiveness of the embeddings may explain the lack of observed gains, but we cannot give a definitive conclusion. Future work could explore injecting more informative or richer representations, as supported by prior literature [cite that paper here].
>
> ## [W3] For the new observations, such as the extension of the ablation study, could they also be able to generalize to other similar models or different datasets outside of those used in the original study?
>
> We extended the original ablation study by including the Google dataset, which was not analyzed in the original XRec paper. The consistent trends we observed across Amazon, Yelp, and Google suggest that our findings may generalize to other datasets with similar characteristics. While we have not tested on additional models, we believe some of the insights, particularly regarding the role of the Collaborative Information Adapter, could be relevant to other LLM-based explainable recommendation frameworks.

---

### Review · Reviewer_EzUY · 2025-04-17

**Summary Of Contributions:**

This paper reproduces and expands upon the findings of XRec, a framework that integrates collaborative signals—patterns of user interactions such as reviews—into Large Language Models (LLMs) by incorporating them into both the input prompt and the model layers. The authors further explore XRec by investigating the role of collaborative signals through the evaluation of different Graph Neural Network (GNN)-based recommender systems, such as LightGCN and NGCF, that capture these signals from graph structures. To assess the impact of entirely removing graph-based collaborative information, the authors conducted three comparative experiments. The conclusion drawn is that the GNN component does not enhance explainability. Instead, the observed performance improvement is attributed to the Collaborative Information Adapter, which acts as a form of soft prompting, efficiently encoding task-specific information.

In summary, this reproducibility paper provides a novel perspective on XRec, demonstrating that removing the GNN-based recommendation system has minimal impact on model explainability, as performance remains stable or slightly improves without it.

**Audience:**

Yes

**Broader Impact Concerns:**

This paper does not raise any ethical implications.

**Claims And Evidence:**

Yes

**Requested Changes:**

1. Correct some proprietary nouns, e.g., Llama -> LLaMA
2. It would be beneficial to explain metrics such as BERT_f1, BERT_recall in the Appendix.
3. More explicitly display the Limitations and Future Work.
4. Additional contributions beyond reproducibility would be advantageous.

**Strengths And Weaknesses:**

Strengths:
1. Revisiting XRec provides evidence that the GNN component does not enhance explainability. Instead, the observed performance improvement is attributed to the Collaborative Information Adapter, which acts as a form of soft prompting, efficiently encoding task-specific information.
2. The experiments are comprehensive, not only providing extensive validation and ablation studies on the original XRec but also offering expanded experiments.
3. The paper is well-written, clear, and understandable.
4. The code is open-sourced and well-documented, facilitating reproducibility.

Weaknesses:
1. The experimental conclusions require further analysis. For instance, injecting collaborative signals into all attention layers of the LLM did not significantly enhance performance, contradicting XRec's hypothesis. The role of collaborative signals in the explainability of recommender systems needs more investigation.
3. The paper's primary contribution is the reproducibility of XRec; other contributions are limited. Proposing better metrics for explainability or a more comprehensive method would make revisiting XRec more solid.

---

> ### Author Response · Authors · 2025-04-30
> **Review response**
>
> Thank you for your thoughtful review. We have addressed your requested changes in the updated manuscript, the changes are highlighted with teal-colored text. We address your weakness points below:
>
> ## [W1] The experimental conclusions require further analysis. For instance, injecting collaborative signals into all attention layers of the LLM did not significantly enhance performance, contradicting XRec's hypothesis. The role of collaborative signals in the explainability of recommender systems needs more investigation.
>
> We appreciate your comment regarding the need for further analysis of our experimental findings. In this paper, we focused on verifying the reproducibility and robustness of claims made in the original XRec paper. To this end, we addressed a critical issue-data leakage-that was not previously acknowledged, and we completed ablation studies that had been left open, particularly involving the GNN architecture.
> While we agree that understanding the role of collaborative signals in explainability deserves deeper investigation, our current goal was to rigorously test the foundational assumptions of the model rather than propose new explanatory methods. We plan to pursue a more comprehensive study of architectural contributions and their role in explainability in future work, where the emphasis will shift from reproducibility to innovation.
>
> ## [W2] The paper's primary contribution is the reproducibility of XRec; other contributions are limited. Proposing better metrics for explainability or a more comprehensive method would make revisiting XRec more solid.
>
> We agree that a stronger contribution could involve proposing new models or metrics for explainability. However, we believe reproducibility is a necessary first step before building on prior architectures. Our work demonstrates that some of XRec's core claims-particularly the performance benefit of collaborative signal injection-do not consistently hold when data leakage is corrected and all components are rigorously tested. Additionally, we provide insight into the impact of the GNN component .
> We also replaced GPTScore with LLMScore, an open-source alternative, making our study more transparent and reproducible. Our findings suggest that architectural changes based on intuition or isolated evaluations should be treated with caution. We hope this work lays a solid foundation for future, more comprehensive advances in explainable recommendation.

---

### Decision · Action_Editor_pRBA · 2025-05-29

**Recommendation:** Accept with minor revision

**Comment:**

This paper re-examines XRec, a popular recommendation model using GNNs and LLMs. It investigates several aspects, including reproducing experiments,  ablation study,  addressing data leakage and etc. The reviewers all agree that the paper has made insightful observations and will benefit the community of recommender systems. For final version, the authors are requested to add discussions on how to develop better metrics for explainability and/or directions for a more comprehensive method that would make revisiting XRec more robust.

**Audience:**

The paper would be of interest to those who work on recommender systems.

**Claims And Evidence:**

The claims in the paper are well-supported with experiments. The code is publicly available for reproducing the results.

---

> ### Author Response · Authors · 2025-06-11
> **Response to decision**
>
> Dear Action Editor and Reviewers,
>
> Thank you for the helpful feedback. In the camera-ready version, Section 5 has been updated to include concrete suggestions for improved explainability metrics, along with the reasoning behind them. Additionally, in Appendix E.1, we have further clarified the computation of corpus-level uncertainty.
>
> We appreciate your time and thoughtful feedback, which have greatly improved our submission.
>
> Kind regards,
> Authors